# Exploring Twitter Discourse around the Use of Artificial Intelligence to Advance Agricultural Sustainability

Catherine E. Sanders *, Kennedy A. Mayfield-Smith and Alexa J. Lamm

Department of Agricultural Leadership, Education, and Communication, College of Agricultural and Environmental Sciences, University of Georgia, Athens, GA 30606, USA; kennedy.mayfieldsmith@uga.edu (K.A.M.-S.); alamm@uga.edu (A.J.L.)
* Correspondence: catherine.dobbins@uga.edu

**Abstract:** This paper presents an exploration of public discourse surrounding the use of artificial intelligence (AI) in agriculture, specifically related to precision agriculture techniques. (1) Advancements in the use of AI have increased its implementation in the agricultural sector, often framed as a sustainable solution for feeding a growing global population. However, lessons learned from previous agricultural innovations indicate that new technologies may face public scrutiny and suspicion, limiting the dissemination of the innovation. Using systems thinking approaches can help to improve the development and dissemination of agricultural innovations and limit the unintended consequences of innovations within society. (2) To analyze the current discourse surrounding AI in agriculture, a content analysis was conducted on Twitter using Meltwater to select tweets with specific reach and engagement. (3) Seven themes resulted from the analysis: precision agriculture and digital technology innovation; transformation and the future of agriculture; accelerate solutions, solve challenges; data management and accessibility; transforming crop management, prioritizing adoption; and AI and sustainability. (4) The discourse on AI in agriculture on Twitter was overwhelmingly positive, failing to account for the potential drawbacks or limits of the innovation. This paper examines the limits of the current communication and outreach across environmental, economic, social, cultural, political, and behavioral contexts.

**Keywords:** artificial intelligence; systems thinking; Twitter; social media; discourse; sustainable innovations



## 1. Introduction

The global agricultural sector faces major challenges, such as a rapid, growing global population, climate change, resource depletion, soil degradation, water pollution, and biodiversity loss [1,2]. There are increasing calls for sustainable agricultural intensification to increase productivity [1], minimize environmental impacts, and provide social benefits [3]. Finding solutions that are economically, politically, environmentally, socially, and culturally sustainable is a seemingly insurmountable challenge; yet, digital technologies are often positioned as the transformational solution for solving global agricultural challenges [4]. The use of digital technologies to transform agri-food systems is often referred to as the fourth agricultural revolution and is characterized by "high-tech, radical, and potentially game-changing technologies" [2]. The digitalization of agricultural systems is aimed at the technological optimization of production, value chains, and food systems, as well as minimizing the environmental impacts of agriculture [5,6]. Artificial intelligence (AI) is one innovation emerging from the digitalization trend, often being used for precision agriculture and to enhance smart farming techniques [6].

AI refers to the development and implementation of intelligent machines or softwares that act by recognizing and responding to their environments, thereby allowing for the analysis of large amounts of data [7,8]. Advancements in AI have already been applied to agricultural production contexts, with predictors expecting they will assist in ensuring

global food security [8], ushering in a wave of AI innovation in the agricultural sector [9,10]. The benefits of integrating AI into agricultural production abound, ranging from improving the traceability of food-related outbreaks, improving hygiene on production sites, integrating supply chains, reducing waste related to production, reducing the agricultural sector's carbon and ecological footprint, and increasing economic profitability [9]. The motivation underlying AI adoption in agriculture is the idea of using innovation to optimize production systems to feed an increasing global population while simultaneously conserving per capita agricultural land area and preserving soil health and environmental quality [6,11].

The role of AI in developing a sustainable agri-food system has been recognized [9]; yet, the holistic social and political sustainability of technology adoption, specifically related to digital technologies, remains a relatively unexplored subject area [3]. According to O'Connor [12], there are four spheres of sustainability: the environmental, economic, political, and social spheres, each interacting and acting upon the others in a reciprocal manner. Research related to AI in agriculture has widely integrated the environmental [13] and economic realm [14], with research emerging targeting the political and social realms [5]. Research to date has primarily focused on the technical aspects of applying technologies to improve agricultural practices or identifying the barriers to AI adoption [5,15,16]. Less frequently mentioned in the discourse surrounding the digital agricultural revolution are the potential negative impacts, or unintended consequences, of digital transformation on economic, environmental, social, and institutional systems that comprise the agri-food sector and beyond [4]. Research into the social sustainability of AI in agri-food systems [3,5,17] foreshadows a need to avoid the consequences of the agricultural innovations of the past, such as those experienced with the introduction of genetically modified organisms (GMOs) [18]. In addition, anticipating unintended public and policy-related challenges emerging from the disruptions ushered in by agricultural innovations [19], especially those that reduce the labor market (which AI does), must be studied [7]. The current research on AI, outside of the agricultural sector, delimits many of the potential pitfalls of AI, including impacts on the labor market, the political landscape, and the medical field [20–22]. Limited research, however, examines the communication and outreach of AI as an emerging technology. Without examining the mechanisms of communication surrounding AI across sectors, scientists risk the efficacy and sustainability of an innovation within the global population. Effective communication and scientific outreach efforts should incorporate two-way communication between scientists and the public to not only improve the reputation of specific innovations but also to improve the development of innovations through listening to audiences and tailoring innovations to their needs [23].

### 1.1. Transformative Technology versus Techno-Optimism

The theory of diffusion of innovations [24], which seeks to explain how and why innovations spread within society, is one of the most widespread theories used in conjunction with agricultural innovation research [25]. The adoption model presented by Rogers [24] in his theory has influenced the dissemination of agricultural innovations across the globe for decades. Rogers emphasized that an innovation should be diffused and adopted among all members of a social system rapidly and with few alterations to the innovation itself. This implication is known as pro-innovation bias. Rogers recognized the inherent critiques to his theory, citing pro-innovation bias, and stating that the spread of new ideas and innovations often yield widening socioeconomic gaps among the members of the targeted social system, often through the unintended consequences of an innovation. Unintended consequences of an innovation occur when technology created with the intent to fix a problem ends up creating new problems resulting from the diffusion and implementation of the innovation [19].

Pro-innovation bias is present throughout the discourse surrounding new agricultural digitization technologies and is often referred to as 'techno-optimism' [26]. A techno-optimism perspective to agricultural innovation often "overlooks social factors, tries to solve social problems with technical fixes, and ignores the unintended negative conse-

quences of new technologies" [3]. Media and policy artifacts may portray emerging technologies in a positive light, perpetuating a techno-optimism perspective that may ignore potential negative on-farm and social consequences of innovations [3]. Discourse surrounding precision agricultural technologies, including AI, frame the innovation as "game-changing" and poised to "transform food production" [2]. Additionally, digitized agricultural innovations are often touted as beneficial to agricultural productivity and the environment, thereby deprioritizing attention to the social consequences of the innovation [3]. This is not to say that precision agriculture technologies, and, in particular AI, do not have the potential for increased productivity and environmental sustainability. Instead, communications surrounding emerging innovations should be positioned as a bridge between scientists and the public to help facilitate discourse around the potential unintended consequences of a technology in order to improve the development and dissemination process of an innovation while minimizing the risk.

Barrett and Rose [3] conducted a literature review exploring the emerging research associated with agricultural technologies, finding the studies most often discussed improved productivity, specifically attuned to the purpose of feeding a growing population. Around half of the articles in their review discussed how the new technologies will improve the environment. Less than half discussed the social impacts of the innovations, and, when discussed, only focused on improving "food traceability, diets, public acceptance, safety, food security, or employment opportunities in the farming sector" [3]. Social issues from an equality perspective were rarely discussed, though there were several scholars focusing their research in this area [2,4,17,27].

## 1.2. Current Discourse

The current agricultural technologies discourse presents digitization and automation as a mechanism for boosting food production and minimizing environmental degradation [28]. Pausing to assess the current discourse around precision agriculture and the use of AI in agriculture is merited to account for both positive and negative impacts given the established patterns of portraying innovations as transformative to the agricultural industry [28]. Policymakers, media, and academic scholars currently frame the fourth agricultural revolution positively, failing to account for the potential negative consequences of innovative technologies and the transformation of agriculture [3]. Despite several previous technologies, such as GMOs, being greeted with enthusiasm by the mainstream media and academics, public controversy ran rampant [28].

Relying only on techno-optimism as a frame for agricultural innovation communication limits an innovation's resilience in the face of negative public opinion and risks perpetuating already-present societal inequalities [28]. While digital technologies help to address divergent global challenges, a focus on techno-optimistic solutions as the primary driver to address wicked problems may exclude responses that are not technology-based when a multi-pronged solution may be more appropriate [2]. Thus, identifying the benefits and drawbacks of emerging agricultural technologies and letting these inform the development and dissemination of innovations may increase their efficacy and resilience within the public sphere [28]. Therefore, analyzing current discourse surrounding agricultural technologies as they emerge may help to identify blind spots that can be addressed through proper communication and outreach and even inform subsequent iterations and creative uses for the innovation. One way to anticipate the unintended consequences of an innovation is through engaging in systems thinking [29].

## 1.3. Systems Thinking Approach

Systems thinking can be used as a tool for thinking through the potential consequences of an innovation. Sustainably governing agricultural innovations requires comprehensively understanding every component of an innovation and how it interacts across cultural and societal levels [30]. The unintended consequences of previous technological revolutions have demonstrated that agricultural innovations are not inherently positive and value-free

endeavors; rather, the success or failure of an innovation is driven by worldviews and diverse visions of the future [4], limiting the social sustainability of the innovation over time [31].

For example, looking at the resistance to GMOs, specifically in the United States and Europe, scientists and communicators can observe the pitfalls that occurred, which may help to predict the acceptance and adoption of similar technological advancements within agri-food systems [18], such as CRISPR and AI. Even if scientists and industry developers perceive their work with agricultural innovations to be environmentally friendly and humanitarian, this does not immediately translate to the broader public.

Systems thinking is a term that is constantly defined and redefined in the literature. Systems are defined as a regularly interacting or interdependent group forming a unified entity, a whole that is greater than the sum of its parts [32]. Systems thinking is a "way to make sense of a complex system that gives attention to exploring the relationships, boundaries and perspectives in a system. It is a mental framework that helps us to become better problem solvers" [33]. Additionally, systems thinking is "a set of synergistic analytic skills used to improve the capability of identifying and understanding systems, predicting their behaviors, and devising modifications to them in order to produce desired effects" [34]. From a systems thinking perspective, the individual parts of a system act differently when viewed in isolation from other component parts of a system [35]; thus, practitioners should explore the inter-relationships of a problem within the social, cultural, environmental, educational, political, and economic perspectives of a system [36] (Figure 1).

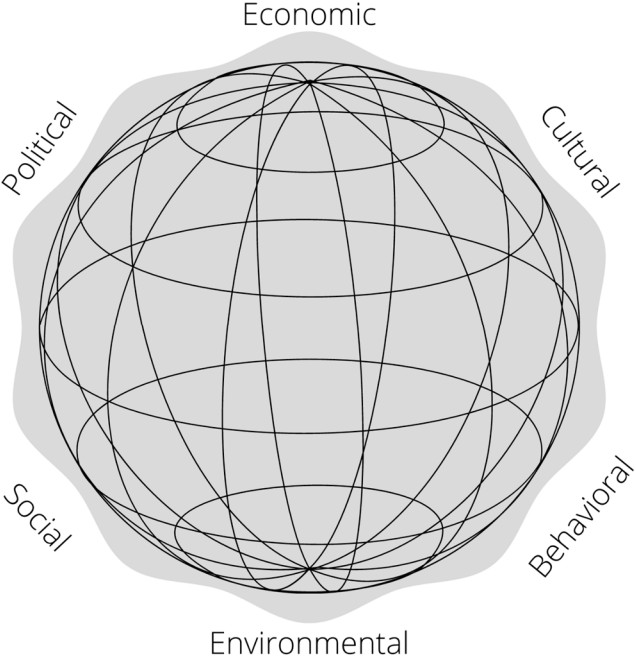

**Figure 1.** Elements of systems thinking.

Within the environmental system, AI is positioned to increase the sustainability of production agriculture [9]. Using a systems thinking approach, however, stakeholders can begin to anticipate the potential unintended consequences of AI [24]. For example, from an economic perspective, AI may radically change the labor landscape within the agricultural sector. Over the past century, agricultural automation has reduced the number of people employed in the sector, thereby reducing employment costs [17] and displacing workers who may not have a transferable skill set. In addition, while automation and digitization do stand to provide beneficial economic impacts to farmers in higher-income nations with access to relevant resources, the political impacts of AI in agriculture would include potentially rising inequalities between farmers in the Global North compared

to lower-income countries in the Global South, exacerbating global equity issues [17]. Implementing AI in farming operations may also radically change the cultural patterns in a community. For example, if being a farmer means becoming skilled in deploying and supervising robotics in the field, this may significantly disrupt the cultural narratives within rural communities [37], impacting the social and psychological well-being of these communities [17]. Additional equity issues that may arise include discrimination against those who are not digitally literate, which could impact rural labor demands and displace marginalized groups currently working in the agricultural sector [38–40].

By using mental frameworks to anticipate the potential consequences of AI adoption in agriculture around the world, scientists may be better informed of the impact of their work outside of strictly technological, natural, and environmental realms. The ability to communicate about AI effectively may assist in bridging communication gaps between the scientists creating new systems and the general public, possibly reducing potentially negative social impacts [18]. However, to determine the existing communication gaps and address them effectively, a baseline assessment of the discourse surrounding AI in agriculture must be conducted. Once the baseline assessment determines the current communication trajectories, a systems thinking framework can be applied to target messaging (and innovation creation) that protects against potential unintended consequences.

## 2. Materials and Methods

The purpose of this study was to describe current discourse around the use of AI in agriculture. Two research questions guided the study: (1) how is AI related to production agriculture currently discussed on social media?; and (2) how does the communication of AI in agriculture relate to sustainable development?

The current study utilized a qualitative research design using social media analysis. Social media analysis is the practice of gathering data from social media platforms and analyzing the data to help researchers address specific problems in relation to the research questions [41,42]. Analyzing social media sources to map policy issues has become popular when striving to determine the direction of policy-related discourse across various societal groups [3,43]. Twitter is a microblogging social media platform, considered to be one of the most popular social media sites across the globe [44]. Twitter is often used as a tool for communication and information exchange relating to a specific topic or issue among users. Twitter was selected as the social media channel for this study due to users' ability to express opinions and highlight concerns surrounding an issue in a unique way when compared to other public spheres and channels [44]. Specifically, a qualitative inductive content analysis of tweets on AI using Meltwater was conducted. Meltwater is a media monitoring and business intelligence software that tracks conversations that people are having on various social media platforms all over the world. The inductive content analysis involved identifying and creating categories from the data based on the research questions and theoretical framework [45]. During the analysis, codes and subcodes were identified that represented different categories. These codes and subcodes were then translated to themes and subthemes.

Using Meltwater, a search was conducted using the Boolean query "Artificial Intelligence" AND (Agriculture OR Farming OR "Agricultural science" OR Ag OR ag). Then, a custom dashboard utilizing the Boolean query was created. The dashboard consisted of the following five widgets designed specifically for media monitoring: trending themes, content stream, media exposure, top location, and top posters by volume. The trending themes widget showed the current number of discussions surrounding and related to the Boolean query. The content stream provided relevant tweets within a certain date range that match the specifications of the Boolean search. The media exposure widget provided insight into how media coverage for a brand, product, event, or topic specified in the Boolean search was trending over time. Identifying the top locations and top posters by volume allowed researchers to gain a holistic view of possible key players in current AI

conversations in addition to the places where conversations surrounding AI in agriculture are on the rise.

Tweets containing or associated with the query "Artificial Intelligence" AND (Agriculture OR Farming OR "Agricultural science" OR Ag OR ag) from 1 January–31 December 2020 were collected and analyzed. We selected the year 2020 because the novel coronavirus (COVID-19) alerted scientists, professionals, and the public to the sensitivity of the global food system, and AI has been a suggested solution to global agricultural challenges [9,46].

Ten tweets were collected for each month, which translated to a total of 120 tweets through a purposive sampling technique to collect 120 of the tweets with highest reach throughout 2020. Purposive sampling is used in qualitative research to identify and select cases that would be most informative for investigating the phenomenon of interest [47]. Tweets were selected based on highest potential reach. Tweets from each handle were copied and pasted into word documents. The word documents were then uploaded to MAXQDA (VERBI GmbH, Berlin, Germany), a data analysis software. The data were analyzed through an inductive thematic analysis approach by two independent coders, in which themes were generated through the identification of repeated concepts or content in the data. Prior to analysis, an intercoder reliability test was run using tweets from the month of September. Cohen's kappa was calculated for the analysis of 10 tweets ($\kappa = 0.68$) and was deemed adequate to move forward with independent coding [48]. The use of a codebook, interrater reliability, an audit trail, and peer debriefing enhanced the trustworthiness of the data [49]. After reading the tweets, a codebook was developed with input from both coders based on the research objectives and theoretical framework [50], providing the foundation for the interpretation of the themes and subthemes.

## 3. Results

Meltwater provided key data analytics to contextualize tweets between 1 January–31 December 2020. Figure 2 provides a heat map of where a large concentration of the tweets occurred geographically. The majority of the tweets were created by users in the United States and India.

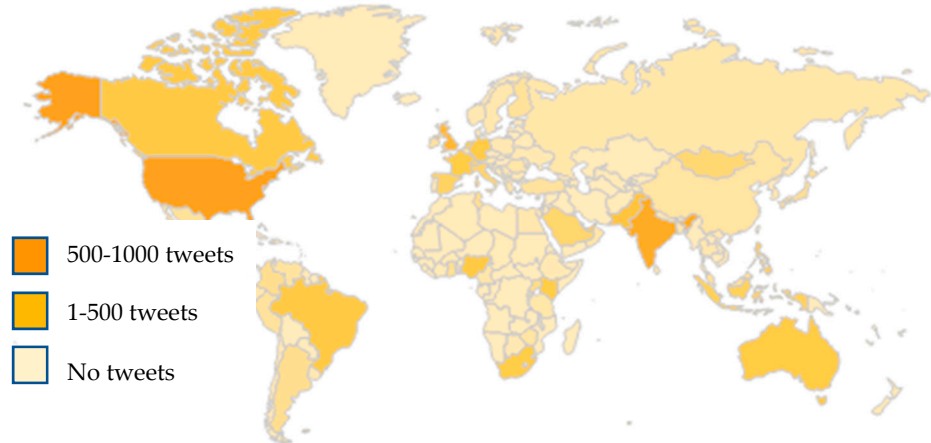

**Figure 2.** Heat map of tweet occurrence from Meltwater.

Meltwater also provided an overview of sentiments toward the topic within the Boolean search query for July–December of 2020. Figure 3 displays the sentiment data throughout the second half of the year, with primarily positive tweets occurring every month except for October, which had equal numbers of sentiment types among the tweets identified by Meltwater.

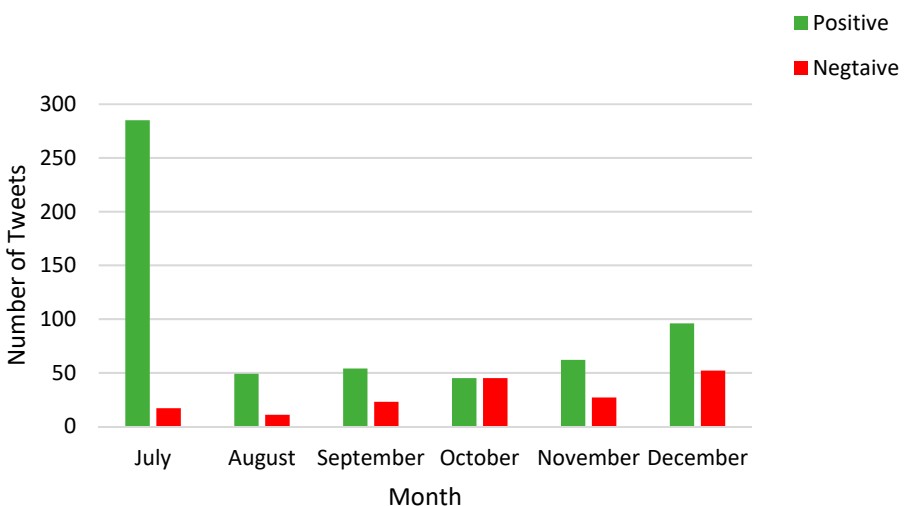

**Figure 3.** Sentiment overview of tweets from Meltwater.

The results are presented as topics or frames used in the discourse surrounding the use of AI in agriculture on Twitter during 2020. The results report was designed to be an overview of the current discourse so that the findings can serve as a baseline assessment and be utilized to inform potential communication strategies and pitfalls moving forward when communicating about innovations and developing research related to AI in agriculture. Table 1 contains the themes and subthemes resulting from the thematic analysis discussed in the subsequent sections.

**Table 1.** Thematic analysis themes and subthemes.

| Theme | Subtheme |
|---|---|
| Precision Agriculture and Digital Technology Innovation | Digitization of agriculture |
|  | Innovations |
|  | Solutions |
| Transformation and the Future of Agriculture |  |
| Accelerate Solutions, Solve Challenges | Feeding a growing population |
|  | Food security |
|  | Climate change |
|  | Higher yields and productivity |
|  | Increasing profits and reducing production costs |
| Data Management and Accessibility |  |
| Transforming Crop Management |  |
| Prioritizing Adoption |  |
| Artificial Intelligence and Sustainability |  |

*3.1. Precision Agriculture and Digital Technology Innovation*

The first theme identified in the analysis was precision agriculture and digital technology innovation. Within the sample tweets, 15 included the digitization of agriculture, primarily discussing blockchain technology, with 28 mentions of precision agriculture. Additionally, of the 54 tweets coded for innovation, two directly mentioned precision agriculture. Within the solutions frame, two tweets directly connected precision agriculture with AI, related to the Precision Ag 201 Webinar (@FarmsNews, November 2020). Additionally, there was an instance of directly connecting AI, precision agriculture, and food sustainability (@UnfoldLabs, July 2020).

There was an emerging discourse directly connecting precision agriculture with AI: "precision farming using artificial intelligence & big data" (@agritechcapital, June 2020). Another tweet framed the rise of AI due to the demands of using precision agriculture techniques: "Artificial intelligence applications in agriculture continue to grow, driven by the increased demands of precision farming" (@UofGCEPS, November 2020). Other tweets implied the future of precision agriculture was dependent on AI, stating "precision #farming involves the usage of innovative artificial intelligence (#AI) technologies for increasing #agriculture productivity" (@SwissCognitive, May 2020). The emerging discourse demonstrates how professionals and scientists in the field believe "artificial intelligence is intertwined in precision agriculture" (@IanLJBrown, December 2020).

### 3.2. Transformation and the Future of Agriculture

Digital technology was a frame frequently used; however, it primarily described how "artificial intelligence will drive #digitaltransformation in agriculture" (@MarshaCollier, 20 January 2020). A frequently used phrase when communicating about digital technologies in agriculture was "help bridge the digital divide" (@IainLJBrown, October 2020) between those with access to and literacy of digital resources and those without, although most tweets discussing innovations viewed them as "transforming" or "transformative" for agriculture" (@2morrowknight, November 2020; @cybersecboardrm, May 2020; @plant_scope, March 2020). Within the transformative frame, AI was positioned to "completely chang[e] the future of farming-giving Ag more sustainability, higher yields, better quality & less loss. It's transforming the Ag industry!" (@mlamons1, March 2020). Innovations in agriculture, including AI, were mostly viewed as "cutting-edge technology" (@detroitnews, February 2020) that will be "beneficial in the agriculture sector" (@DDNewslive, January 2020). Others urged governments to "bring digital technology-led revolution in [the agricultural] sector" (@MinisterKTR, August 2020). Terms like "revolutionizing" were common in this digitization discourse (@CristinaAtFpc, 27 August 2020), as well as how digital technology is "redefining" agriculture (@IainLJBrown, November 2020).

### 3.3. Accelerate Solutions, Solve Challenges

The solutions and challenges included feeding a growing population, food security, climate change, higher yields and productivity, increasing profits and reducing production costs, and crop management. Specifically, most solutions from AI in agriculture were positioned to "tackle global challenges" (@CSIRO, June 2020) and make the agricultural industry "more efficient and environmentally friendly" (@GREATBritain, April 2020). Another tweet boasted how AI in agriculture already "solved a farming challenge" (@CSIRO, August 2020). While infrequent, references were made to AI helping "ensur[e] social good" (@abhish18, February 2020).

Solutions for feeding a growing population was a popular frame for discussing AI in agriculture, highlighting AI "and its role in ensuring food for all" (@FAO, October 2020). Additionally, food security frames were similarly invoked, emphasizing how AI can "improve food security and eradicate hunger" (@DailyNewsZim, October 2020). Several tweets combined solutions for food security with those of climate change, making these frames highly interrelated, such as "can we use artificial intelligence to solve climate change and feed the growing world? You can't spell climate or agriculture without AI" (@techreview, October 2020), and how AI can help solve "some of the world's biggest challenges, including food security and climate change" (@BradSmi, 24 September 2020). Additionally, the FAO was connected to this work, hosting webinars to discuss how "technology can help unlock solutions for some of the world's biggest challenges, including food insecurity and climate change . . . AI can help support and accelerate the @FAO's important work" (@BradSmi, September 2020). With climate change specifically, AI was cited as "helping to breed crops for the changing climate" (@PNASNews, October 2020).

An additional frame for discussing AI in agriculture was how AI could produce higher yields and/or increase productivity. This frame was often coupled with a need "to produce

more with fewer resources" (@plant_scope, March 2020). Tweets pointed to a history of using technology to improve agriculture: "Farming has regularly used technology to improve yields & the industry is now looking at adopting Artificial Intelligence in many ways including analyzing crops to better manage yield" (@mlamons1, May 2020). Specifically, AI in agriculture could help "study the soil and the optimal cultivation conditions to improve its yield" (@VinitalyTour, June 2020).

Increasing profits was another frame touting the benefits of AI in agriculture, describing how AI would make "farming more profitable" (@GREATBritain, April 2020). Often, this frame was coupled with the frame of reducing production costs, in which "Artificial intelligence equipped machines will also play a big part in the future of agriculture, reducing food production costs and improving land use" (@BernardMarr, September 2020). Lower production costs were associated with improved crop management practices: "#Agriculture Industry Moves Forward Using Artificial Intelligence (#AI) To Improve Crop Management" (@TamaraMcCleary, May 2020).

### 3.4. Data Management and Accessibility

Technical discourse surrounding how AI would help with data management and accessibility was also present. One tweet described new "harvest loss analysis technology" (@jicksonstephen, February 2020), while another demonstrated how AI could "doubl[e] farmers' income on open agriculture data for India" (@ISBedu, March 2020). AI was positioned as advancing precision agriculture through big data (@agritechcapital, June 2020). Growers would also "now [be] able to directly access information about products including product details and dosage" (@Syngenta, August 2020).

### 3.5. Transforming Crop Management

In addition to revolutionizing on-farm data management, how AI in agriculture was revolutionizing farmers' capacity for crop management was also discussed. AI was described as being able to "predict corn yield rates [with] precision agriculture" (@into_AI, March 2020) and "detect crop diseases" (@DavidPraiseKalu, April 2020). Other capacity developments included "creat[ing] an autonomous strawberry picker that does the job twice as fast as humans" (@BernardMarr, September 2020) and a "a weeding bot–an automated robot using artificial intelligence to identify and remove weeds from rows of crops" (@Kenyans, December 2020).

### 3.6. Prioritizing Adoption

Aligned with Rogers' [24] diffusion of innovation theory, prioritizing the adoption of AI and precision agricultural practices was another prevalent theme. Tweets described organizational-level priorities for advancing adoption: "The Food and Agriculture Organisation says the government must fast-track the adoption and use of digital tools and artificial intelligence to improve food security and eradicate hunger" (@DailyNewsZim, October 2020). Other tweets described the "Rising Adoption of AI-Enabled Devices in Agribusiness" (@IainLJBrown, October 2020). AI was depicted as a national priority for India across sectors, including agriculture (@NITIAayog, 2020).

### 3.7. Artificial Intelligence and Sustainability

Several tweets pointed to the nexus of using digital agricultural technologies, such as AI and precision agriculture, as a means of increasing agricultural sustainability. General connections echoed the following: "giving [agriculture] more sustainability, higher yields, better quality & less loss" (@mlamons, March 2020) and "aim[ing] to promote sustainable farming through artificial intelligence and machine learning" (@PCMag, October 2020). Other sustainable impacts included "producing food that is potentially more affordable and more sustainable" (@Botanygeek, October 2020). More specific sustainable impacts included "tackl[ing] global challenges including illegal fishing and plastic waste" (@CSIRO, June 2020) and "creating resilient farm/food systems" (@agritechcapital, June 2020). The

President of Pakistan was quoted as connecting water conservation and AI: "'The pricing of water shall add to water conservation. Agriculture productivity is essential so is the inclusion of artificial intelligence', said the President" (@NAofPakistan, March 2020). He expanded his view of the anticipated sustainable impacts of AI when he stated, "Agriculture is the backbone of our country [Pakistan] and the intervention of artificial intelligence is a must to upgrade the conventional farming methods" (@AsadQaiserPTI, March 2020). Other examples of sustainability and AI had more micro-applications, such as managing vertical farms in order to use "99% less land" (@nigewillson, January 2020).

## 4. Discussion

The findings supported previous research that described the discourse around agricultural innovations as overwhelmingly positive [2,3]. While the data generated from Meltwater depicted a majority of positive tweets with a subset of negative sentiment tweets in the total analytics for latter half of 2020, the data set analyzed through the discourse analysis only analyzed tweets with the largest reach; thus, the results indicate that, while there were negative sentiment tweets on the social media platform, those with the highest reach and engagement had a positive sentiment, which drives the public discourse on Twitter. The general discourse within precision agriculture, and the AI conversation specifically, is operating under a pro-innovation bias [24], supported by the findings presented in the current study. Maintaining a techno-optimistic stance in outreach alone does not ensure widespread adoption in the public sphere, which will ultimately impact policy for a specific innovation if it trends negatively, as evidenced by the pitfalls experienced related to GMOs [18,28].

Using systems thinking to consider the consequences of an innovation within social, political, cultural, economic, and environmental contexts (Figure 1) may be an effective tool in making improvements to the innovation itself prior to widespread dissemination [30]. With the predominately positive outlook on AI in agriculture, few themes specifically addressed the potential drawbacks of the innovation, revealing a lack of systems thinking practice within innovation development and dissemination. Taking all aspects of a system adopting AI into account can also be used to improve outreach and science communication efforts related to AI as the industry considers widespread adoption as a solution to broad agricultural issues [24]. Considering the results, the environmental implications of AI in agriculture are already embedded within Twitter discourse, specifically as identified in the themes and subthemes of climate change, higher yields and productivity, and AI and sustainability. The use of AI in agriculture was positioned as a necessary solution for combating climate change, enhancing the environmental sustainability of production agriculture, and reducing land use through higher yields. The sustainability frame of AI in agriculture may be an effective messaging strategy when attempting to influence policy among more environmentally focused decision-makers, as well as environmentally focused potential adopters of the innovations.

The economic context was also present within Twitter discourse, depicted through the increasing profits and reducing production costs theme. The economic frames, however, were predominantly production-oriented, limiting the messaging effects for policy related to increasing the use of AI and precision agriculture techniques. Economic arguments, both on the producer and the national/global scale related to market value, can increase buy-in for policymakers intending to promote more environmentally sustainable agricultural practices. Combining the environmental and economic contexts in the messaging can broaden the potential scope and efficacy of the innovation as these messaging strategies can demonstrate an environmentally focused solution with potential economic benefits. An additional gap in the economic discourse on Twitter was accounting for the potential of job loss within the agricultural sector due to AI and the associated anxiety of this loss in the labor market [17]. Without these discussions taking place within the public sphere (in this case, on social media), scientific innovations risk significant backlash and suspicion

in the economic, social, and political contexts of the innovation both on the national and global scale [17].

The frames oriented toward the social context of AI in agriculture primarily focused on social challenges, such as feeding a growing population and food security. The social frames were primarily discussed in a global context, relating to complex challenges at the nexus of climate change and pressures on the food system. However, AI has the potential to disrupt the social patterns in farming communities, relating not only to the social context but the political and cultural contexts as well. For example, the need for digital literacy and access could create unequal power structures within social systems where AI is implemented, especially in a global setting [38–40], so strategies for outreach should be attuned to minimizing this risk considering these contexts simultaneously.

Within the behavioral context, communications surrounding the diffusion of AI in agriculture remain centered around prioritizing adoption, aligned with Rogers [24] diffusion of innovation theory. However, limited messaging strategies incorporated other aspects of behavior change related to AI aside from the general benefits of adoption. In order to promote the social sustainability of AI in agriculture among not only producers, the messages should target policymakers and others with decision-making power to outline the needed actions for the sustainable integration of AI in agriculture across the diverse agri-food system. The contexts in which innovations exist continually overlap at the interstices of social, cultural, political, economic, environmental, and behavioral systems; thus, the potential unintended consequences of an innovation should not be discussed within an isolated context. Through the systems thinking approach, stakeholders are encouraged to anticipate the ripple effects of an innovation across these six contexts, increasing the sustainability of the innovation across the system.

Relative to the higher-level AI conversation, the discourse around the agricultural applications of AI is a small subset of the general AI discourse on Twitter. Remaining secluded from the broader AI communication spectrum, while potentially positive now due to the ability to avoid broader controversies attributed to AI, the agricultural sector is extremely susceptible to any backlash should a crisis occur or public opinion quickly turn. As evidenced in the results, the agricultural industry sits at a precipice of choosing to be proactive in obtaining public support for the use of AI to increase the sustainability of agriculture or remaining dependent upon blind techno-optimism that could limit the innovation's resilience [28].

The findings indicate that the practitioners working at the intersection of precision agriculture and sustainability have a unique opportunity to be proactive communicators and strive to build a relationship and connection to the public around AI's use in agriculture. Connecting sustainability and precision agriculture in outreach efforts, by placing an emphasis on environmental sustainability, the public trust in production agriculture can be improved, especially with AI exhibiting the potential to have a massive global impact on the labor and food market [2]. Specifically, social media communication campaigns should be developed that highlight the benefits of AI use in agriculture using visual imagery and video that the public can associate with solving environmental issues (e.g., reduced leaching into lakes and rivers, reduction in algae blooms in popular recreational areas, increased yields with less fertilizer and pesticide application). The communication campaigns could be collaborative, where scientists and agricultural companies and/or farmers are showcased working together to identify and evaluate the benefits of AI use. Once in place, the impact of the campaign should be measured, and compared to the baseline shared here, to determine if the public discourse is altered by the communication effort. Logically conducting public discourse analysis over time will ultimately help to predict backlash against any innovation that may be introduced, and, when combined with the systems thinking approach, improve the system-level sustainability of innovations prior to dissemination.

## 5. Conclusions

The overall discourse around AI and precision agriculture was generally positive, without widespread consideration of the potential drawbacks of the innovation, which supported previous research [2,3]. The implications of the study include emphasizing a systems thinking approach for both innovation development and dissemination to improve the system-level of an innovation. The framework depicted in Figure 1 may assist as a reference point to ensure practitioners and scientists consider the various contexts in which the innovation will interact, emphasizing the need for considering the social, cultural, political, environmental, economic, and behavioral aspects of an innovation. With many livelihoods depending on the agricultural sector, combined with a broader need for increased sustainable practices, a systems thinking approach to innovation development and dissemination will combine critical thinking with evidence-based science to enhance sustainability across sectors.

The limitations of the current study include only examining agricultural AI discourse from one social media site, Twitter. Future research should examine emerging discourse across social media sites, such as Facebook, YouTube, Instagram, or TikTok, to compare how discourses emerge based on the communication channel. Additionally, future research should explore discourse emerging through news articles as several tweets in the current study referenced news articles that impacted the users' perceptions of the innovation. While the current study focused only on Twitter as a baseline assessment of emerging discourse, future studies that examine more communication outlets can provide a broader description of the agricultural AI discourse both through qualitative and quantitative methodologies.

**Author Contributions:** Conceptualization, C.E.S. and A.J.L.; methodology, C.E.S., K.A.M.-S., A.J.L.; software, C.E.S. and K.A.M.-S.; formal analysis, C.E.S., and K.A.M.-S.; investigation, C.E.S. and K.A.M.-S.; resources, C.E.S., K.A.M.-S., and A.J.L.; data curation, C.E.S., and K.A.M.-S.; writing—original draft preparation, C.E.S., and K.A.M.-S.; writing—review and editing, C.E.S. and A.J.L.; visualization, C.E.S. and A.J.L.; supervision, A.J.L.; funding acquisition, A.J.L. All authors have read and agreed to the published version of the manuscript.

**Funding:** This work was supported by the USDA National Institute of Food and Agriculture, Hatch project 1021735. Any opinions, findings, conclusions, or recommendations expressed in this publication are those of the authors and do not necessarily reflect the view of the National Institute of Food and Agriculture or the United States Department of Agriculture.

**Institutional Review Board Statement:** Not applicable.

**Informed Consent Statement:** Not applicable.

**Data Availability Statement:** Data presented in the study were pulled from publicly available data. The authors pulled data from Twitter, a public-use platform, to code for qualitative analysis. The coded data is available upon request.

**Conflicts of Interest:** The authors declare no conflict of interest. The funders had no role in the design of the study; in the collection, analyses, or interpretation of data; in the writing of the manuscript, or in the decision to publish the results.

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
