# Peer review of "Exploring Twitter Discourse around the Use of Artificial Intelligence to Advance Agricultural Sustainability"

_sustainability, doi:10.3390/su132112033_

Round 1

Reviewer 1 Report

I think the article is interesting, although I missed the need for a "conclusion" section. Therefore, I think it might be useful to include some conclusions with contributions to theory and practice, along with limitations and suggestions for future research.

Author Response

The authors have provided a specific conclusion section, briefly summarizing the findings in relation to theory. We also expanded the implications of the study to more specifically advocate for a systems-thinking framework to improve the sustainability of innovations across sectors. Additionally, we added a more direct limitations section, which we followed with future research recommendations based on these identified limitations.

Reviewer 2 Report

It s an interesting study, done according to accepted methodologies, with important conclusions

Author Response

The authors appreciate your feedback on the study.

Reviewer 3 Report

The research carried seems to be have an appropriate findings with respect to two questions framed. The researcher has considered only Twitter as social media platform to carry this work. This could have been better, if the data could have been collected from different social media platforms. Any how, I appreciate if the authors can address these few points in their paper.

  1. The Research on AI in agriculture on Twitter responses were all positive? If not, Need to mention the Negative discussions? To reflect the findings realistic and also strengthening the discussion. if any.
  2. Need to provide the graphical represenatation of statistics to number of tweets positive/negative received in total also with respect to period.
  3. What were the objectives and theoritical framework used for interpretation of the themes and subthemes. They need to be included in the paper to support the work.
  4. Why did the analyist filtered/considered only ten tweets for each month? Why is this threshold value considered?
  5. After identifying the themes and subthemes, the authors should have conducted survey to validate and to identify the findings are appropriate or not? 

Author Response

While the scope of the current study specifically focused on Twitter due to its structure as a microblogging platform, the authors have accounted for this limitation in expanding future research and describing how future research could explore multiple social media sites. Since this was an exploration of emerging discourse, we selected one platform as a baseline assessment with the hopes that future research could expand upon these findings.

Responding to points 1 and 2, we have added sentiment data from Meltwater to contextualize our data. We have also addressed the implications of the sentiment data from Meltwater with our qualitative analysis, explaining that the tweets with most reach and engagement have a positive sentiment, with negative tweets not having as high of reach.

Responding to point 3, themes were analyzed inductively from the data. We have updated the methods to be more specific about the thematic analysis approach. We did not use a deductive approach to thematic analysis, which would have directly used a priori themes developed from the theoretical frameworks and objectives. The role of the theoretical framework was to connect frames/themes identified in the data to reveal potential drawbacks and trajectories of current discourse.

Responding to point 4, the authors have described their motivations for the purposive sampling technique used to select cases most informative for the phenomenon of interest. This sampling technique selected 10 tweets with the highest reach resulting in 120 cases for qualitative analysis across the year examined.

Responding to point 5, since we conducted a qualitative thematic analysis related to discourse of AI in agriculture on Twitter, we enhanced the trustworthiness of our data analysis through interrater reliability and the kappa coefficient. Conducting a survey to validate themes is not required for this type of analysis; rather, through interrater reliability, an audit trail, and peer debriefing, we enhanced the trustworthiness of our data in accordance with Lincoln and Guba’s 1985 recommendations (in their book Naturalistic Inquiry, citation and clarification added to methods section).

Round 2

Reviewer 3 Report

The respective comments are addressed in the current revision